# Dermoscopic image segmentation based on Pyramid Residual Attention Module

**Yun Jiang**◎*, **Tongtong Cheng**◎*, **Jinkun Dong**◎, **Jing Liang**◎, **Yuan Zhang, Xin Lin, Huixia Yao**

College of Computer Science and Engineering, Lanzhou, Gansu, China

◎ These authors contributed equally to this work.
* jiangyun@nwnu.edu.cn (YJ); 2020211957@nwnu.edu.cn (TC)

## Abstract

We propose a stacked convolutional neural network incorporating a novel and efficient pyramid residual attention (PRA) module for the task of automatic segmentation of dermoscopic images. Precise segmentation is a significant and challenging step for computer-aided diagnosis technology in skin lesion diagnosis and treatment. The proposed PRA has the following characteristics: First, we concentrate on three widely used modules in the PRA. The purpose of the pyramid structure is to extract the feature information of the lesion area at different scales, the residual means is aimed to ensure the efficiency of model training, and the attention mechanism is used to screen effective features maps. Thanks to the PRA, our network can still obtain precise boundary information that distinguishes healthy skin from diseased areas for the blurred lesion areas. Secondly, the proposed PRA can increase the segmentation ability of a single module for lesion regions through efficient stacking. The third, we incorporate the idea of encoder-decoder into the architecture of the overall network. Compared with the traditional networks, we divide the segmentation procedure into three levels and construct the pyramid residual attention network (PRAN). The shallow layer mainly processes spatial information, the middle layer refines both spatial and semantic information, and the deep layer intensively learns semantic information. The basic module of PRAN is PRA, which is enough to ensure the efficiency of the three-layer architecture network. We extensively evaluate our method on ISIC2017 and ISIC2018 datasets. The experimental results demonstrate that PRAN can obtain better segmentation performance comparable to state-of-the-art deep learning models under the same experiment environment conditions.

## 1. Introduction

Skin cancer is approximately one of the most cancer cases. Currently, two to three million cases are increasing worldwide each year. According to the American Cancer Society, there were 108, 420 new cases of skin cancer and 11, 480 deaths in 2020, an increase of 12.37% and 58.78% year-on-year [1]. The lethality rate of skin cancer can reduce by early diagnosis. There is still 95% curative rate in the early stages of the disease in malignant melanoma even. But

**Data Availability Statement:** The relevant data needed to replicate this study are available in the GitHub repository: https://github.com/SixCorePeach/PRANet.

**Funding:** This work was supported by National Natural Science Foundation of China in the form of a grant (61962054) to YJ. This work was also supported by The Cultivation Plan of Major Scientific Research Projects of Northwest Normal in the form of a grant (NWNU-LKZD2021-06) to YJ. The funders had no role in study design, data collection and analysis, decision to publish, or preparation of the manuscript.

**Competing interests:** The authors have declared that no competing interests exist.

once it worsens, the survival rate of patients is only 15% [2]. The early diagnosis of skin lesions is indispensable because malignant melanoma has concealment and long latency. The dermoscopy is designed to provide the physician with high-resolution images of abnormal parts of the patient's epidermis. And physician can further identify the symptoms by segmenting the images manually and reducing the probability of misdiagnosis [3]. However, it is still a challenging task to achieve the accurate segmentation of these images. As shown in Fig 1, dermoscopic images exist with low contrast, a non-uniform appearance, and a large number of interfering factors (such as hair, artifacts, artificial markers, etc.), which obstruct the diagnostic process [4]. Computer-aided diagnosis (CAD) technology plays an essential role in solving similar challenges. After delineating the lesion area from the normal skin area through instruments accurately, physicians can identify the diagnostic area directly and improve the accuracy of the diagnosis [5].

In recent years, deep learning methods have been used increasingly in computer vision, natural language processing, and medical image process. Convolutional neural networks(CNNs) have achieved good results on many tasks of medical images segmentation. Compared with traditional manual segmentation methods, by learning the feature distribution in the dataset and adjusting the parameters only from the image itself, CNNs can obtain faster, better, more stable segmentation results. A few years ago, PSPNet [6] with a pyramid pooling module (PPM) could fuse features from multiple sensory fields and improve the learning ability of the model for data with complex and diverse features types. Wei et al. [7] proposed to put the PPM on each skip connection of U-Net [8] for improving the segmentation of the model and used a pixel attention mechanism to weigh the results of the pyramid process increasing in the perceptual field of PPM. Recently, the emerged attention mechanism can effectively extract global information from the image. It can provide references for local information extraction, focus on the distinctive parts of the image, and eliminate irrelevant information, thus achieving more accurate segmentation results. As the first segmentation model combined attention in tackling the medical image processing tasks, Attention-U-Net [9] applied a spatial attention-like mechanism to control the information at skip connections. It supports filtering out relevant information and suppressing the transmission of irrelevance. The SE-block proposed in Squeeze-and-Excitation networks [10] can obtain the rank of each channel in feature maps

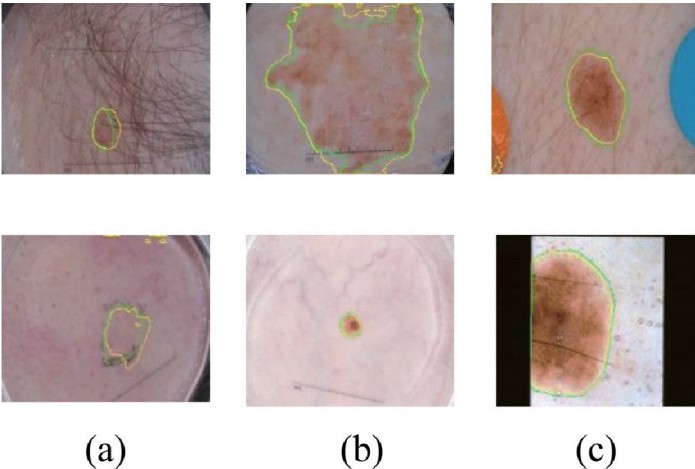

(a)                      (b)                      (c)

**Fig 1.** (a) Background differences between skin and lesions with poor contrast. (b) Irregularity and variable size of lesions. (c) Images of lesions are often accompanied by a large amount of noise. The green line represents Ground truth and the yellow is segmented by our model.

by global average pooling, and achieve the weight assignment of feature channels better. CA-Net [11], ADAM [12] incorporated bidirectional, channel, spatial, and multi-scale attention mechanisms into the encoder-decoder structure. One or more combinations of attention mechanisms were aimed to capture spatial information from a shallow layer thereby enriching the underlying semantic information. CA-Net placed the attention module on the up-sampling and skip connection based on U-Net. An adaptive bi-directional attention module proposed in ADAM supplemented the shallow extraction results with the attention information at the bottom level. Although the above attention mechanisms can improve the segmentation effect of neural networks to different degrees, they mostly contain the following limitations:

1. The single convolution kernel has a limited ability to learn various features. The different sizes of the receptive fields have an important impact on the feature extraction process. For feature-diverse datasets, the feature extraction ability of the model can be enhanced further by the multi-scale feature fusion mechanism.

2. The importance of the convolution layers has been ignorant for each stage during the design of the network framework. It is reasonable to adopt various convolution strategies when confronted with different sizes of semantic information with different densities. Deeper sub-networks can be assigned to sufficiently extract messages in the shallow stage with comprehensive information.

3. It is not screened for the information processing of feature extraction in classic networks. There is ignorant of the different situations of the amount of information extracted by the convolution kernels in the feature extraction modules. The module which is responsible for extracting information should be supervised and managed accordingly, so as to enhance the update iteration of important convolution kernels and reduce the unnecessary training cost.

To address the above shortcomings, we propose a novel pyramid residual attention(PRA) module, assemble a network PRAN (Pyramid Residual Attention Network) with good segmentation effectiveness. Compared with existing work, it uses the PRA module consisting of feature pyramids, residual units, and attention as the base unit of the network to ensure the stability and robustness of the model. In this paper, our work can be summarized in the following points:

• We propose a novel pyramidal residual attention(PRA) module. It consists of several basic convolution structures used widely in current neural networks. This module can automatically extract features of different sizes from the original image, with good feature extraction ability for skin cancer images.

• Differing from the traditional U-Net-based network, this paper proposes a network composed of combined PRA modules—PRAN. We divide the traditional encoder-decoder structure into three stages. Every stage in PRAN is aimed to learn the feature maps of different sizes for the enhancement of the encoder-decoder. These designs alleviate the information loss of the U-shaped network due to excessive sampling to a greater extent.

• Our PRAN achieved good segmentation results on both the ISIC2017 dataset and ISIC2018 dataset.

The rest of this paper is arranged as follows: Section 2 introduces the related works. Section 3 introduces the proposed module and network. Section 4 presents the experimental details. Section 5 gives our conclusion in the end.

## 2. Related works

In this section, we mainly introduce four accesses for skin lesion segmentation tasks, namely attention mechanisms, residual and pyramid attention networks, loss functions and preprocessing methods.

### 2.1. Attention mechanisms

Starting from U-Net and FCN [13], there were many attention mechanisms proposed, such as the channel attention [14], the spatial attention [15], the multi-scale attention [16], the cross attention [17], and the bidirectional attention [18], etc. Atrous convolution and pyramidal pooling structure also participated in the process of improving the model's ability to process complex objects. He et al. [19] fused multi-input RefineNet to the residual structure. Wang et al. [20] proposed setting each small convolution kernel in a spatial pyramid with attention. Shahin et al. [21] combined pyramidal pooling with U-shaped networks by placing the pyramid pooling structure on the skip connections of U-shaped networks to learn the spatial information from the encoder. Kaul et al. [22] designed FoucsNet to combine channel attention with two stacked U-Net and introduced the idea of coarse and fine segmentation. Ding et al. [23] proposed a fusion method of depth-aware gated fusion mechanism with U-Net, which used the gating mechanism to enhance the extraction of the encoder part. Xie et al. [24] reconciled two encoder-decoder structures with attention-graph structure. They used attention based on the graph to process the result of the coarse segmentation and deliveries an input to the fine segmentation. Gu et al. [11] also combined U-Net with kinds of attentions, adding improved spatial attention on the skip connection, putting channel attentions on the up-sampling part, and assigning multi-scale attentions at the end of the multi-outputs. Subsequently, a new adaptive dual attention mechanism [12] was proposed to the skip connection of U-Net to capture context effectively. Sarker et al. [25] proposed a multi-input encoder-decoder network, and Lu et al. [26] complemented a CO-attention mechanism that can enhance the model to capture remote information. In addition, a multi-branch encoder-decoder network [27] applied channel attention, spatial attention, and multi-scale fusion mechanism to extract features from images at each level.

### 2.2. Residual and pyramid attention networks

The pyramid structure and the residual structure are identical to attention in different ways. They are all capable of improving the learning ability of the model to varying degrees. Mei et al. [28] proposed a pyramid attention structure that combined convolution kernels of different sizes with scale agnostic attention to learn the global feature information of an image. Fu et al. [29] designed a residual pyramid attention network for CT image segmentation by combining the inception-like module with SE-block and attention block consisting of a small encoder-decoder, and then combining the two to form a feature extraction module with a large number of parameters. Chae et al. [30] implemented a residual UNet network combined with SE-block for wound region segmentation, and a modified version of SE-block was added to the skip-connected part of UNet to raise the shallow information transfer efficiency. SAR-U-Net [31] also used a combination of SE-Block and the pyramid pooling embedding in the Res-UNet. This composition can increase the ability of feature capture of the downsampling in Res-UNet. In [32], residual units and astrous convolution were regarded as the basic convolution unit and heightened the learning ability of the model on features. Flaute et al. [33] proposed a residual channel attention network for sampling and recovery of super-resolution images, which can well preserve the integrity of features learned from the encoder. Punn et al. [34] designed a residual space cross-attention-guided inception-Unet model that fused shallow

and deep semantic information and improved the extraction capability of a single convolutional block with the inception [35] structure.

## 2.3. Loss functions and preprocessing methods

Starting from the loss function and preprocessing, the former uses a specific loss function to make the model easy to converge, and the latter improves the dataset quality. Yuan et al. [36] first used Jaccard distance as a loss function to effectively deal with the background and focal area imbalance case. Subsequently, Zhang et al. [37] proposed a Kappa loss based on the Kappa coefficient, which takes into account all pixel points of the segmentation result, thus improving the accuracy of the model prediction. Abhishek et al. [38] designed the MCC loss function based on the Mathews correlation coefficient, similar to Kappa loss when taking into account the individual values of the evaluation confusion matrix. Sarker et al. [39] proposed a loss function with a combination of cross-entropy loss and Euclidean distance. Lu et al. [40] implemented the Shrinkage loss function to balance the number of data of the different classes. Saha et al. [41] designed a color enhancement strategy to enrich and expand the dataset by decomposing the image into layers of different hues. Li et al. [42] used a deep learning method as a preprocessing method for hair removal, which has a significant improvement in enhancing the segmentation of the model. Abhishek et al. [43] fused HSV, RGB, and grayscale images to obtain the input shadow decay image, which provides better conditions for accurate segmentation of the model. Since the U-shaped network has the defect of excessive information loss, we use the PRA module as the backbone of the network and make use of the advantages of the encoder-decoder idea to form a combined network. In addition, combining with the advantages of existing work, we use the Dice loss as the loss function to enable the model to learn the features of skin lesion images more effectively and achieve better and more stable results.

## 3. Methods

In this section, we introduce the PRAN (Pyramid Residual Attention Network), the three-layer encoder-decoder network for image segmentation of skin lesions, and the PRA, the module-the basic unit of the PRAN.

### 3.1. Structure of the Pyramid Residual Attention Network (PRAN)

U-shaped networks or U-Net-based neural networks have insufficient ability to extract feature maps of different sizes. The single-kind convolution kernels for information extraction of semantic and spatial information of different densities will lead to the loss of feature information to some degree. So we design the PRAN consisting of a pyramidal residual attention module. Our PRAN consists of a three-layer structure which are the shallow layer, middle layer, and deep layer. Incorporating the idea of encoder-decoder structure, each layer consists of a combination of separate PRA modules. The processing of feature extraction is performed only in the current layer. We choose up-sampling and down-sampling to finish information fusion in different layers and information is reprocessed by various skip-connect operations in the same layer. The PRAN is illustrated in Fig 2, and the parameter distribution is presented in Table 1.

- The shallow layer uses a combined unit of two PRA modules to process the input of the original image specification. The feature output obtained by the first attention mechanism weighting forms a skip connection, which is merged with the result of the second information extraction. Then we use the channel attention module to weigh again and fuse with the

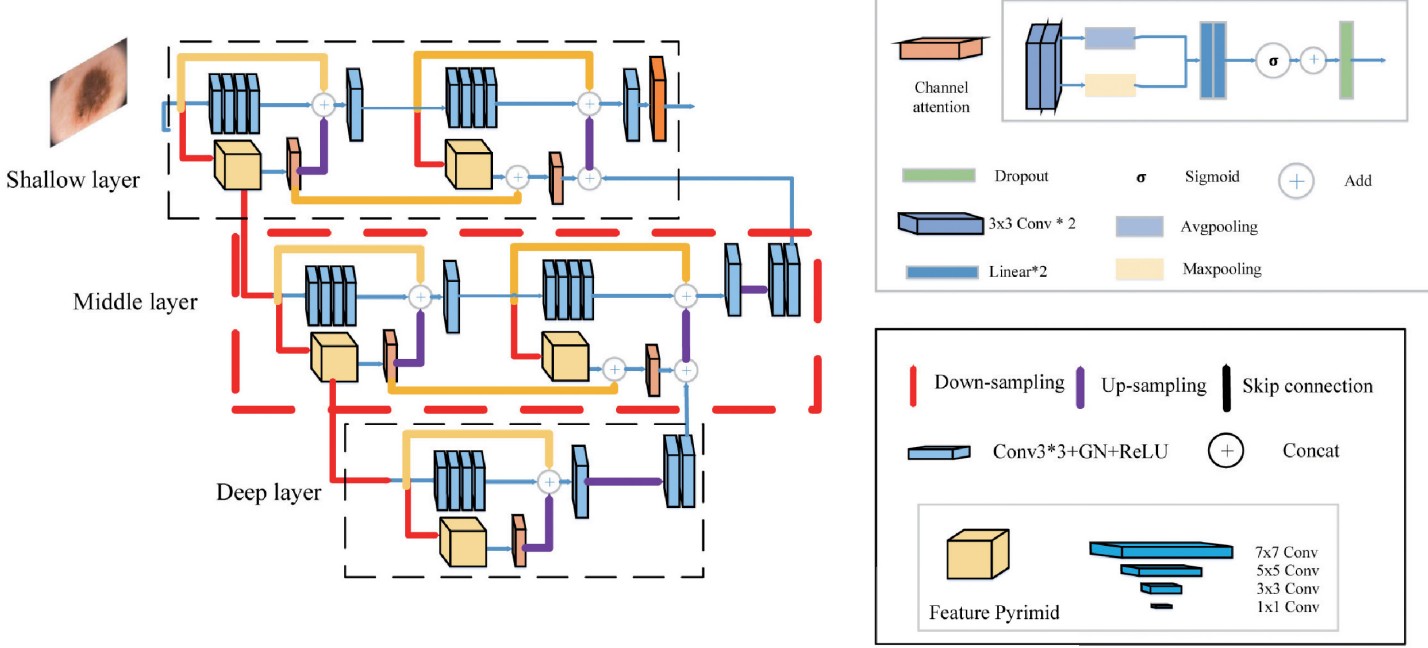

**Fig 2. Pyramid residual attention network.**

information from the middle layer to obtain the final output. The residual unit of the module consists of two simple two-layer 3 × 3 convolutions containing group normalization(GN) [44]. The residual unit retains the initial information while extracting features and waiting for the features from the lower layers to add for the final segmentation.

- The input of the middle layer comes from the feature pyramid of the shallow layer. After a maxpooling layer, the feature maps are 1/4 the size of the original image. It is accompanied by the reduction of spatial information and the condensation of semantic information. The processing of the feature maps in this layer is the same as that in the shallow layer. The output of the middle layer is obtained by supplementing the information extracted from the deep layer.

- There is only one PRA module in the deep layer, yet its role is crucial. After two times of feature pyramid extraction and four times of down-sampling, the resulting feature maps are only 1/16 of the size of the original maps, which contain dense semantic information. The information extraction mechanism at the bottom layer enables the hidden features to be fully learned, complementing information at the middle layer and thus enhancing the learning capability of the model.

**Table 1. List of parameters of PRA modules in each position in the model, where ( ) represents the (1, 3, 5, 7) four convolution kernels with the same channels.**

| Location | Pyramid | Attention | Skip connection | Post process |
|---|---|---|---|---|
| Left shallow layer | 8×( )×( ) | 32×3×3 | 29×3×3 | 64×3×3 |
| Right shallow layer | 16×( )×( ) | 160×3×3 | 64×3×3 | 32×3×3 |
| Left middle layer | 32×( )×( ) | 128×3×3 | 64×3×3 | 64×3×3 |
| Right middle layer | 64×( )×( ) | 448×3×3 | 64×3×3 | 64×3×3 |
| Deeplayer | 128×( )×( ) | 512×3×3 | 128×3×3 | 32×3×3 |

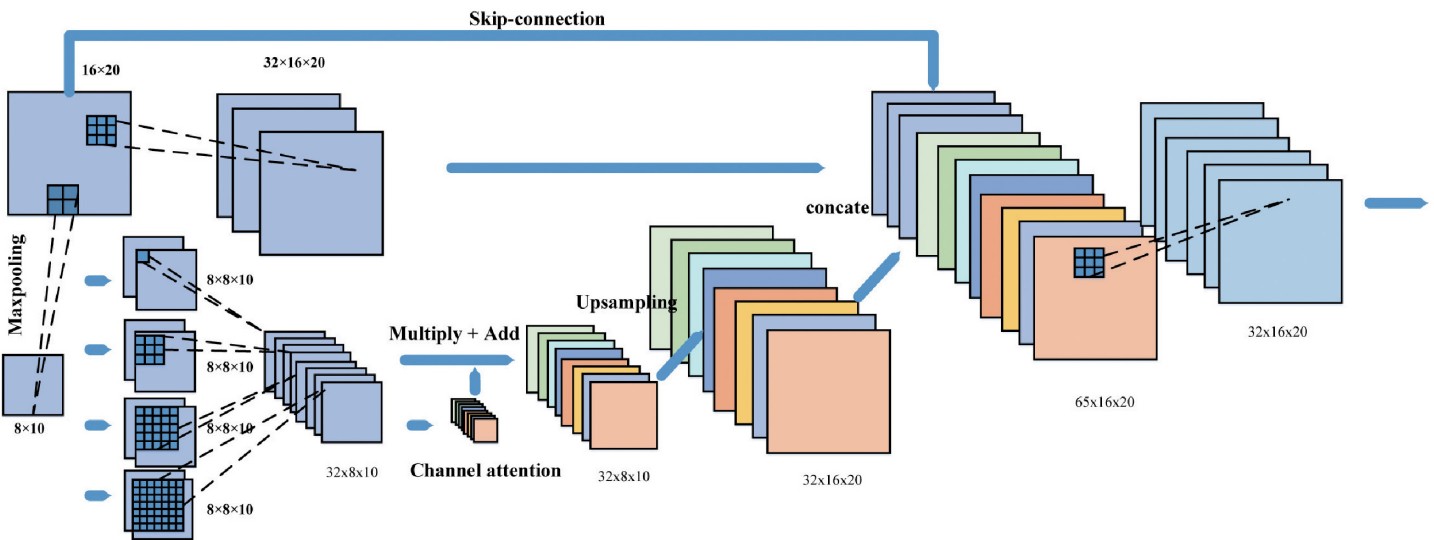

**Fig 3. Pyramid Residual Attention Module.**

## 3.2. Pyramid residual attention module

We propose a streamlined and effective synthesis module called the pyramidal residual attention (PRA) module. To better understand the principle of the module's composition, we illustrate the concept of feature learning with a simple convolutional neural network. Fig 3 shows the flow of the PRA module for processing feature maps. Taking the deep layer of the PRAN as an example, the size of the input image and the output feature map is 16×20. Firstly, the feature maps are processed based on the original maps extracted by 32 convolution kernels size of 3×3. At the same time, the input feature maps are compressed by a 2×2 pooling layer with a step size of 2, concentrating the semantic information and ignoring the useless spatial information. Subsequently, feature maps are fed into a feature pyramid to learn the feature information in different sizes of receptive fields. The feature sets of multiple receivers are used as the input for the channel attention. The channel attention mechanism takes the results of different convolution blocks to derive the information weight of each channel through a pooling layer. These arrangements can enhance the input to various degrees to focus the attention. Finally, after recovering the size by the deconvolution layer, the feature maps connect with the output of the residual block to obtain the output of the module.

**3.2.1. Feature pyramids.** In PRA, feature pyramids are feature extractors that incorporate different kinds of convolutional kernels. Because features of different dimensions can be learned efficiently by these convolution kernels, and the operation of convolution is pixel-by-pixel. We apply four convolutional kernels of 1×1, 3×3, 5×5, and 7×7 to process the input in parallel. Multiple different convolutional kernels can learn sufficient knowledge from different perceptual fields [45]. Different kinds of convolutions can add more semantic and spatial information to the results than the single 3×3 convolutional kernel. The stride of the convolution kernel is set to 1, which does not change the size of the feature extraction so that the features of the image can be substantially preserved and provide information security for subsequent learning.

**3.2.2. Channel attention.** Traditional convolutional neural networks lack a feature screening mechanism for deep network features. Bahdanau et al. [19] proposed an attention mechanism for this situation arising in natural language processing which can effectively

alleviate such problems. Considering skin cancer lesion areas have irregular boundaries, these important edge features need to be learned adequately in different feature sizes. Channel attention is used as a supervised unit to screen features to reduce redundancy. In the original SE-block [10] module, only the global average pooling is used to extract the features of channels, which will make the spatial information lost seriously. So we use the maximum pooling for information supplementation. Specifically, as in Eqs (1)–(4): we define $x$ as the module input, $y$ as the module output, $x \in H \times W \times C$, the global average pooling function $P_{gavg}$, the maxpooling function $P_{gmax}$, among the equation, $P_{gavg}(x) \in R^{1 \times 1 \times C}$, $P_{gmax}(x) \in R^{1 \times 1 \times C}$, $k_1, k_2, b_1, b_2$, in the formula represent the two linear layer weights and biases. The sigmoid function can produce the distribution of weights between channels when the weights rely mainly on both their input and adjustment of the linear layer. Refers to the per-channel multiplication operation, feature maps are first passed through the global average pooling layer and the global maximum pooling layer, thus obtaining important information for each channel.

$$P_{gavg} = \frac{1}{H \times W} \sum_{x=1}^{H} \sum_{y=1}^{W} F^c(x, y) \tag{1}$$

$$P_{gmax} = \arg\max\{F^c(x, y), x \in [0, H], y \in [0, W], c \in [1, C]\} \tag{2}$$

$F^c(x, y)$ represents the values on the c-channel coordinates $(x, y)$ on the feature map. Both global average pooling and maximum pooling screen the importance of each channel and rank them by the sigmoid function.

$$Y = sigmoid(k_2 \times (k_1 \times (P_{gavg}(x) + P_{gmax}(x)) + b_1) + b_2) \tag{3}$$

$$Out = Y \otimes x + x \tag{4}$$

The obtained sequence of the importance of feature maps is channel multiplied with the input and finally summed with the original inputs.

**3.2.3. Residuals mechanism.** As shown in Fig 3, in our PRA, the two branches of the residual structure [46] extract feature maps at different scales, while the skip-connected part passes the input directly backward. The convolution and group normalization [44] can reinforce the learning of the module for small batches of images. The result of the other branch comes from feature pyramids and channel attention. This residual structure can be comprehended as a mapping, as in Eq (5). And there is another mapping in the PRA module. The residual structure used in this paper is shown in Eq (6).

$$F(x) = H(x) - x \tag{5}$$

$$G(x) + F(x) + x = H(x) \tag{6}$$

For the feature extraction part of PRA, if the input feature $x$ is already optimal, the directional gradient propagation in the residual structure can set the gradient to zero, thus speeding up the training processing of the model. The shortcut can ensure that the model will move to be better steadily during the training. When the shallow parameters of the model reach a specific optimal value, the deep parameters will update on top of that and not move in a worse direction. The inclusion of the residual structure not only improves the performance of the model but also speeds up the training of the model and prevents the gradient from disappearing [39].

**3.2.4. Loss function.** Considering that the lesion areas of dermoscopy images are irregular and noisy, we choose Dice loss [47] as the loss function to ensure the steady of model training. Since the Dice coefficient is an important index, using Dice loss can increase the Dice coefficient of the model in a targeted manner when improving the learning ability of the model. As

shown in Eq (7), *P* refers to the output of the network, *G* represents the label of the image. Dice loss is better than IOU loss [36] and Cross-Entropy loss in maintaining the model's generalization in our experiments.

$$Diceloss = 1 - \frac{2 \times P \cap G}{|P| + |G|} \tag{7}$$

## 4. Experimental results

All our methods were implemented on the Pytorch framework, using Adaptive Moment Estimation (Adam) as the training method for gradient descent, while setting the hyperparameters as follows: weight delay was 1e-6, batch size was 8, and the number of iterations set 300. The experience was implemented on a Quadro RTX 6000 GPU (24G), used a stepped learning rate with an initialized learning rate equaling 0.3, and changed it to 1e-4 after 30 iterations and to 1e-6 after 270 iterations. To verify the validity of this model, we conducted experiments using the following three approaches:

1. The model was trained and tested on ISIC2017. The results obtained from the experiments were compared with the advanced models in recent years, and the detailed experimental results are presented in 4.4.

2. We trained the PRA module on the ISIC2018 dataset and demonstrated its effectiveness. We implemented the PRAN, performed ablation experiments on the test set, and compared our model with state-of-the-art models in recent years.

3. To increase the validity verification of the model, the following models were replicated in this paper: U-Net, CA-Net, ATTU-Net, DenseASPP, CE-Net, and the results were presented in Tables 4 and 5.

### 4.1. Data sets and evaluation metrics

For skin lesion segmentation, our PRAN was evaluated on the datasets of ISIC Challenge 2017, 2018 for two years. The ISIC2017 dataset [48] contains a total of 2750 images, of which 2000, 150, 600 are used as the training, validation, test set respectively. In contrast, the ISIC2018 dataset [49] has only the complete training set totaling 2594 images. Therefore, we randomly divided the ISIC2018 dataset into three parts, with 1816 images in the training set, 256 as the validation set, and the remaining 518 as the test set. The ISIC2018 dataset contains data from ISIC2017, so the ISIC2018 dataset is introduced briefly here. The images in the dataset are of different sizes that are RGB images of sizes 720×540 ∼ 6708×4439 are to be normalized as the input to the model before the segmentation task. Here we normalize all the images to the size of 320×256 and enhance the dataset by flipping horizontally and vertically, cutting randomly, and rotating -90 ∼ 90 degrees, as shown in Fig 4.

We used DC(Dice coefficient), MIOU, ACC(accuracy), SP(specificity), and SE(sensitivity) as important metrics to evaluate the performance of the model. Among them, ACC is the ratio of correctly predicted pixel points to all pixel points. Suppose there are many pixel points, *G* represents the pixel points of positive and negative classes in the labels, and *P* represents the positive and negative classes in the prediction results, and *k* denotes the intersection operation of the set, where the Dice coefficient (DC) and MIOU coefficient are operated as

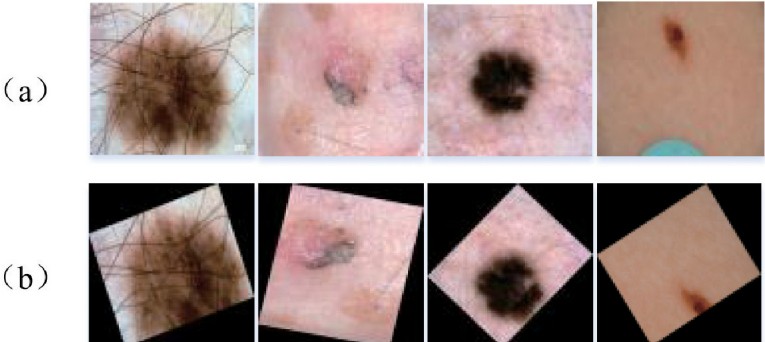

**Fig 4. Image display using data enhancement, (a) original image (b) normalized and data enhanced image.**

Eqs (8) and (9).

$$DC = \frac{2 \times k(G, P)}{G + P} \qquad (8)$$

$$MIOU = \frac{k(G, \bar{P})}{k(G, P) + k(\bar{G}, P) + k(G, \bar{P})} \qquad (9)$$

## 4.2. Ablation experiments for module validation

We tested the performance of the network and verified the rationality of the PRA module through ablation experiments. The effectiveness of individual PRA modules was verified on the ISIC2018 dataset using the same experimental environment and hyperparameters. The feature learning capability of the modules is derived by comparison of the Precision-Recall plot (PR) and receiver operating characteristic curve (ROC) plot in Fig 5. As shown in Table 2, the effect of the feature pyramid, attention mechanism, and residual module on PRA is observed by replacing each widget using a 3×3 convolutional kernel, where the replacement of the attention mechanism causes the performance of the feature extraction module to decrease by approximately 14%, as shown in Table 2 for PRA-CA (Channel Attention), while the replacement of the pyramid PRA-PM (Pyramid Module) only degrades the performance of the model by 1%. The area of the ROC and Precision-Recall curves show that the combination of the three can enhance the sensitivity and specificity of the model to different degrees, and further justifies the module design. Table 3 shows the test results of the combination of PRA modules on the ISIC2018 dataset. We find that a single PRA module can achieve significant performance improvement after horizontal and vertical slicing. Among them, horizontal stitching can improve the learning ability of the modules, and vertical stitching can further learn

**Table 2. Table of experimental results of the effectiveness ablation of each component of PRA module.** The PM, CA, Res stands for Pyramid Module, Channel Attention, Residual Unit, respectively, "-" refers to replace the module with a 3×3 convolution kernel. Bold data indicates the maximum value in this indicator.

| Network | DC(%) | MIOU(%) | SP(%) | ACC(%) | SE(%) | Para(kb) |
|---|---|---|---|---|---|---|
| The PRA—PM | 86.80 | 78.16 | 90.83 | 91.36 | 91.89 | 150.17 |
| The PRA—CA | 75.92 | 64.12 | 82.60 | 83.02 | 83.46 | 102.69 |
| The PRA—Res | 84.93 | 75.69 | 89.58 | 90.08 | 90.60 | 86.26 |
| The PRA(Proposed) | **87.86** | **79.37** | **91.48** | **92.01** | **92.56** | 122.45 |

**Table 3. Validation table for PRA module made up in horizontal and vertical directions.**

| Network | DC(%) | MIOU(%) | SP(%) | ACC(%) | SE(%) | Para(M) |
|---|---|---|---|---|---|---|
| The PRA | 87.86 | 79.37 | 91.48 | 92.01 | 92.56 | 0.1 |
| The Shallow of PRA Network | 91.75 | 85.58 | 94.13 | 94.85 | 95.58 | 1.1 |
| **PRAN(Proposed)** | **93.37** | **87.94** | **94.98** | **95.56** | **96.15** | 21.6 |

features at different scales, thus enhancing the generalization ability of the model for lesion segmentation of rugged skin cancer images. According to the consensus principle of network composition, the generalization ability.

As shown in Fig 6, when facing the segmentation of different types of skin lesion areas, the PRA module can often capture only a small amount of information. There is still the problem of noise interference for the determination of lesion areas, as shown in Fig 6d, while by deepening the model horizontally, a part of the noise can be reduced. Then by deepening the model vertically and assembling the PRA modules into PRAN, the noise removal ability of the model was improved greatly, as shown in Fig 6c, which segmentation effect obtained is also closer to Ground Truth. The different combinations of PRA modules have different enhancements in the ROC curve and Precision-Recall curve.

As shown in Fig 7, the horizontal building improves the sensitivity and specificity more obviously. The vertical three-layer construction increases the accuracy and recall of the model further. The depth and width expansion of the network also accompany the increasing of parameters, which is worthwhile as far as the effect is concerned. We continued deepening network layers in the horizontal direction in the trial experiment. But we found that it did not bring any apparent improvement. Thus we turned to expand the stages of the model, fixed the number of layers of the model at three, and achieved good segmentation results in the end.

As shown in Fig 8, the loss decline curves for training and validation, our proposed PRA module can reach a good level of loss in 50 iterations during training and validation and makes a slight and stable decline in the training loss in the last 250 iterations. Moreover, our PRAN has a more obvious advantage over UNet in line with loss reduction and is more stable.

## 4.3. Comparison with state-of-the-art methods on the ISIC2017 dataset

We find that the virtual height of the model training on the ISIC2017 dataset exists in the invalidation process. The evaluation metric of the generalization performance of the model decreases by 4% ∼ 5% compared to the valid set. From another perspective, this situation is precisely proportional to the feature learning ability of the model. The effect of our model is in Table 4. Through comparing with the existing models, we find that the segmentation of the lesion region is related to the robustness of the model on the ISIC2017 dataset as seen in Fig 9. The effect of the pure channel and spatial attention to enhance the model on this dataset was more limited, for example, CA-Net, DS-Net [44], etc. However, some models like CE-Net [45], ADAM, could achieve good performance gains by enhancing the feature extraction mechanism or the combined attention. As shown in Fig 8, our model can segment different types of lesions satisfactorily with minimal interference from noise compared to the other models. Since there is hardly a distinction between the lesion area and the background, the segmentation results are represented by highlighting the color of the lesion area. We used pink as the base color in the picture. After increasing the hues of pink, the differences between lesions and skin can be seen in the results. By visualizing the results, the physician can recognize the boundaries of the lesion area easier and how the darker parts of the lesion area are highlighted in the segmented area.

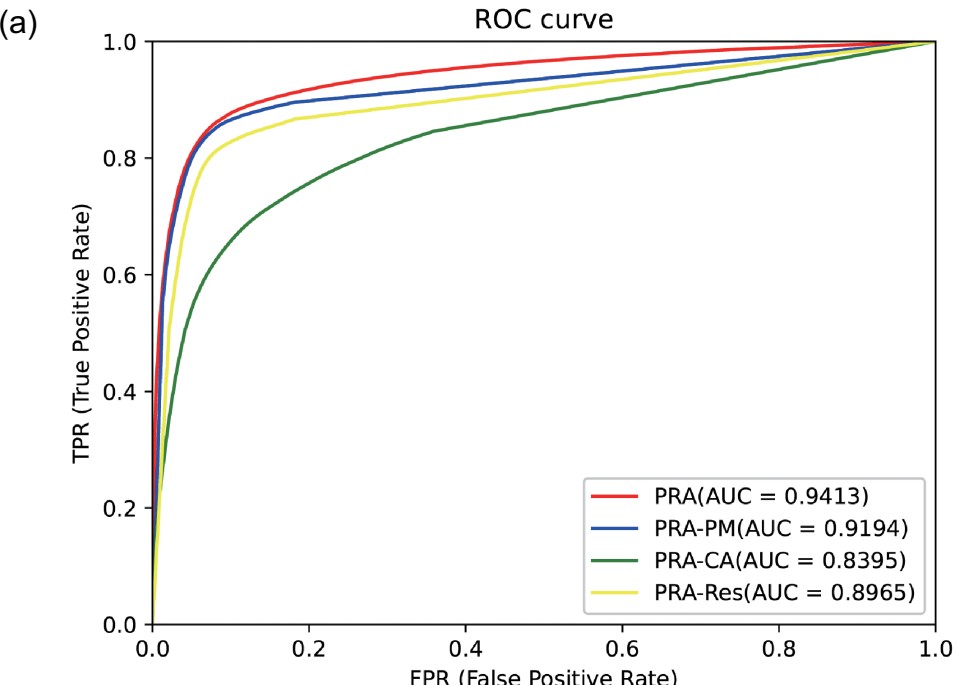

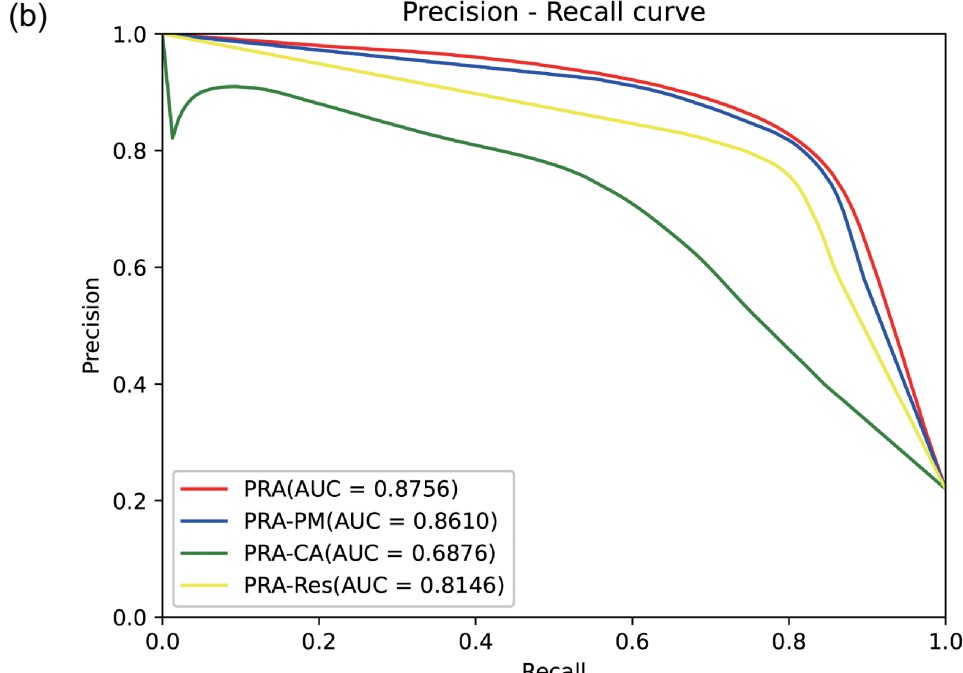

**Fig 5. Precision-Recall curve and ROC curve for the effectiveness of each component of the PRA module.**

## 4.4. Comparison with state-of-the-art methods on the ISIC2018 dataset

The results of the proposed models are compared with the current optimal models as shown in Table 5. We can observe the strengths and weaknesses of the models from the tabulated data.

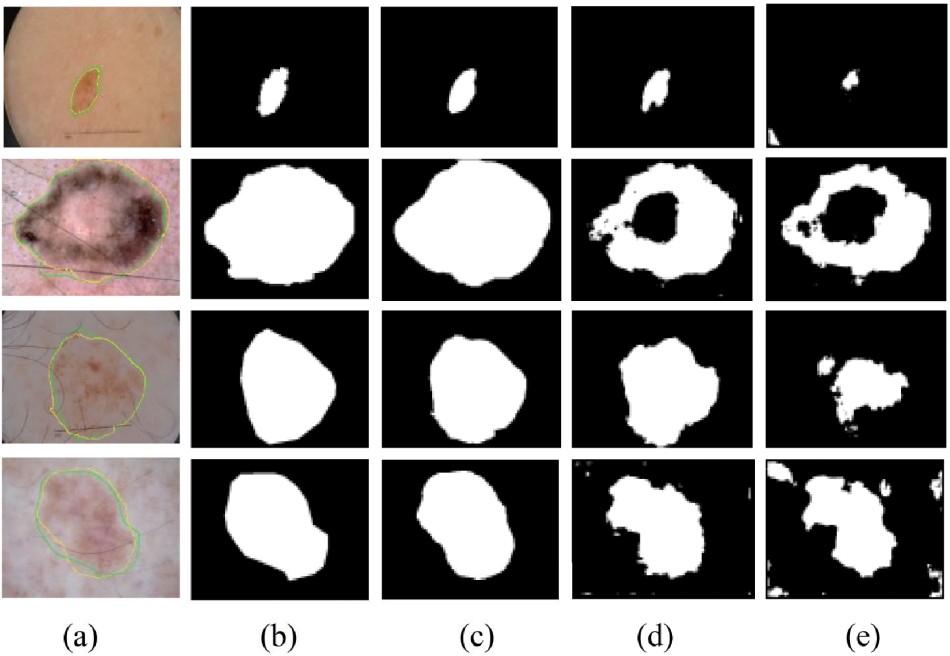

**Fig 6. The segmentation results of the proposed PRA module and PRAN test on the ISIC2018 dataset.** Here, (a) is one of the images, the green line represents Ground Truth, and the yellow line represents the segmentation result of PRAN. (b) shows the labeled graph of Ground Truth, (c), (d), (e) is the segmentation result of PRAN, the horizontal combination of two PRA modules, and a single PRA module, respectively.

The performance of the models is fully demonstrated by the visualization of the segmentation results as shown in Fig 10. The comparison of our model with other models on the ISIC2018 dataset shows that the validation set can affect the training processing of the model in some cases. In addition, on the validation set, the results obtained by the model in those two data sets are not significantly different in each metric. However, the final test set shows that the validation set of the ISIC2018 dataset was more credible. Fortunately, The proposed PRAN achieved good results in the comparison of results. In the segmentation of skin lesion images, our model needs to segment the central region of the lesion with significant contrast and make a reliable guess of the general shape of the lesion. Even when the contrast of images is not high, our model can still ensure that the lesion area is within the segmented region. Through the above ablation experiments and comparative studies, our PRAN can achieve good segmentation results in skin cancer images which consist of diverse backgrounds and oddly shaped lesions. Due to the effective mechanism of the PRA module, it was capable of learning the knowledge sufficiently in the feature maps. The difference from previous networks is that the encoder-decoder idea is no longer limited to the design of modules, but integrated into basic blocks. With the insurmountable deficiency of insufficient feature extraction in U-shaped networks, it was not enough to use various attentions to improve U-Net, but worthwhile to learn and develop the application of the encoder-decoder idea. The basis of the PRA module combination was a continuation of the deep-learning idea that networks with more parameters tend to be more powerful, which is why our PRAN can obtain performance improvements.

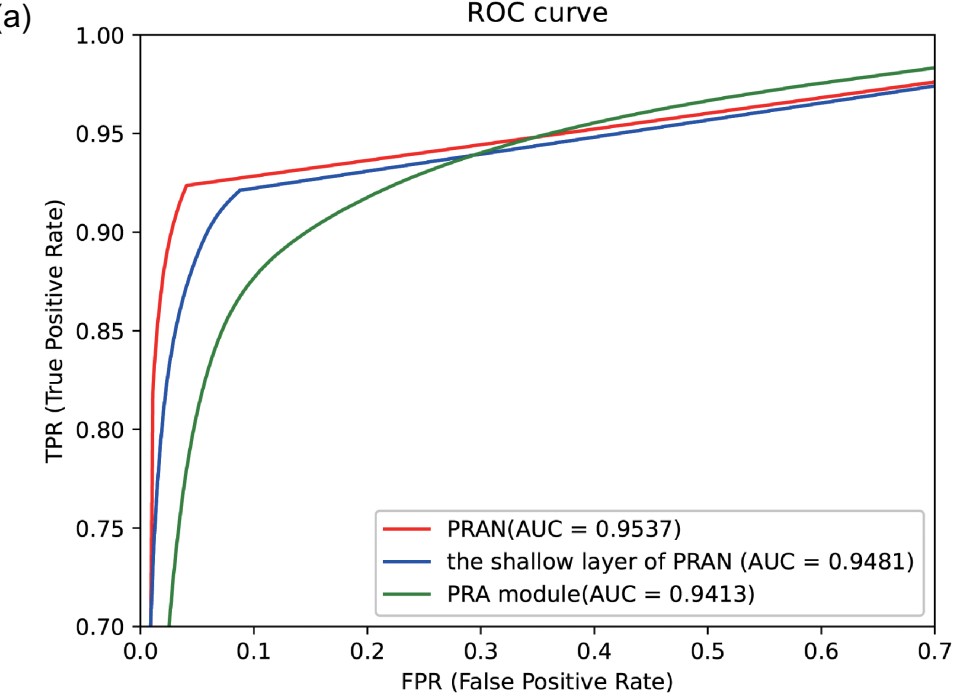

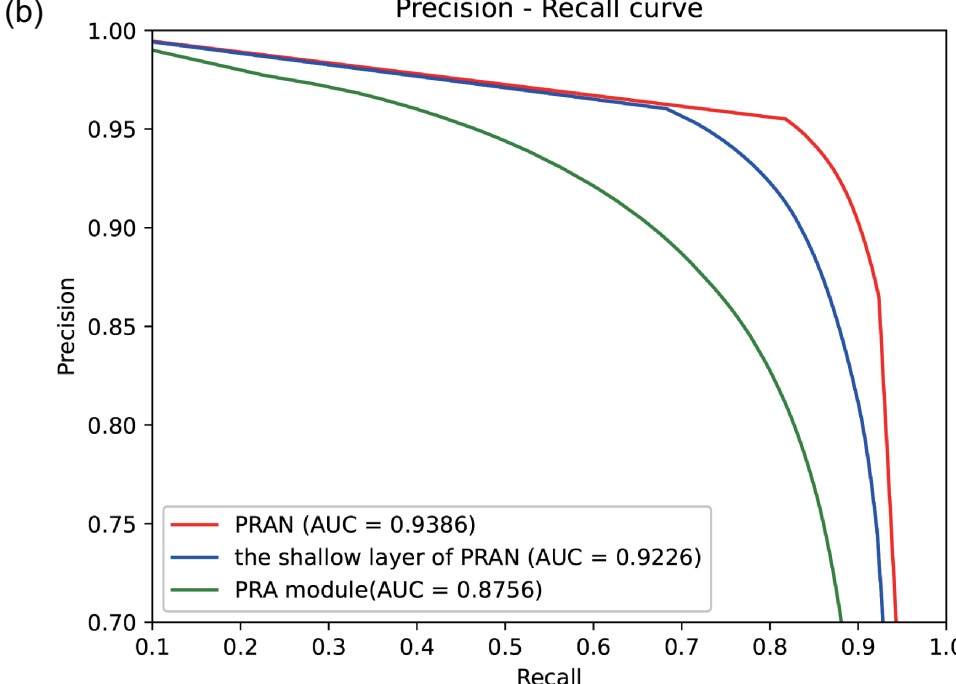

**Fig 7. The Plot of PR curves for the combined ablation experiment, where PRA is a single module, the shallow layer of the PRAN is that has been combined horizontally, and PRAN is the complete network after assembling of PRA modules vertically.**

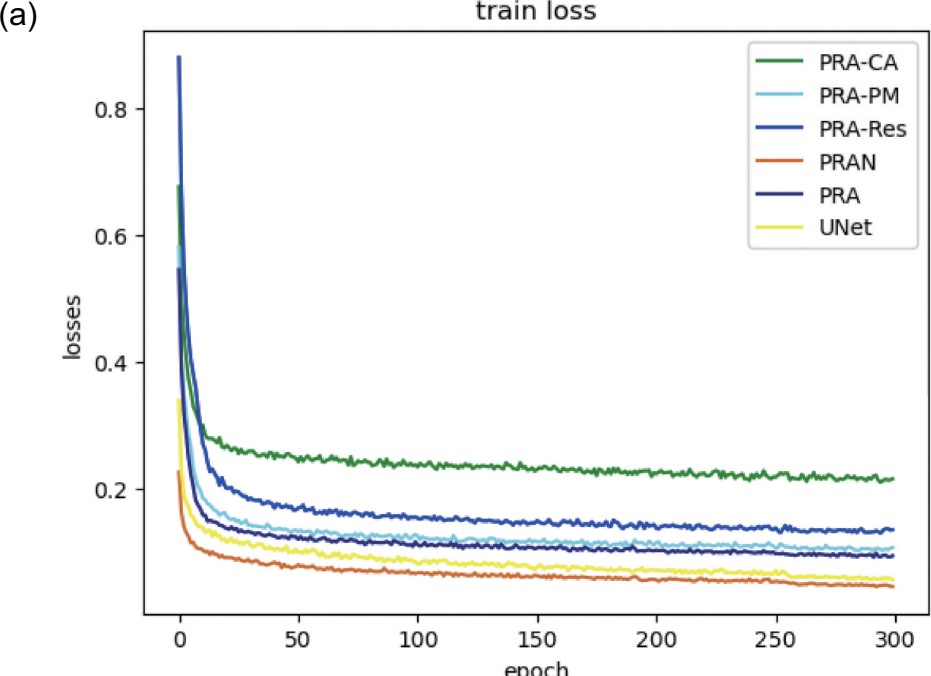

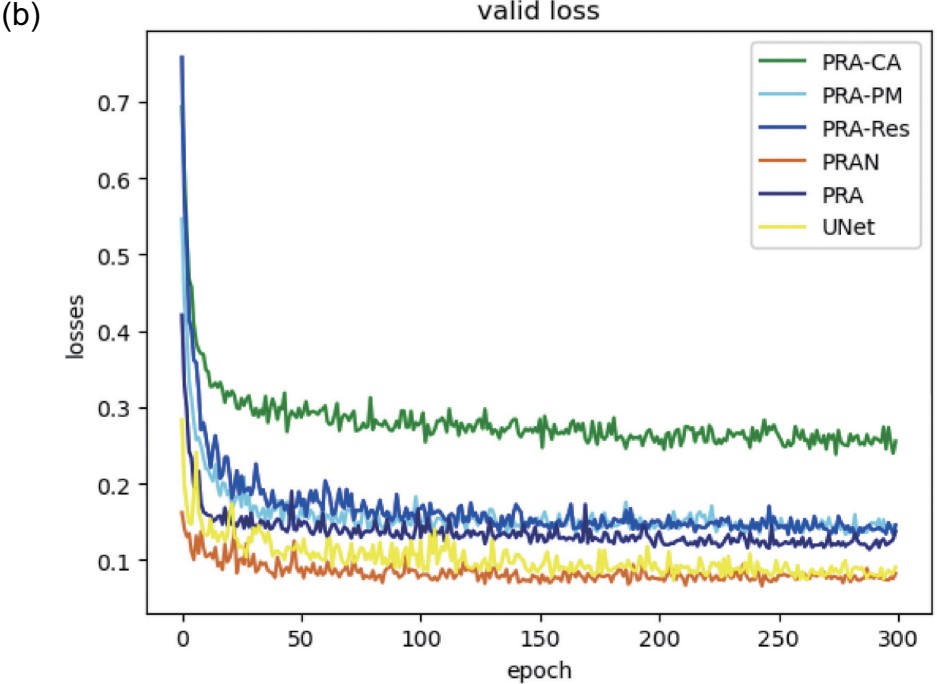

**Fig 8. The Plot of train and valid loss curves for the combined ablation experiment.** PRA is a single module, the shallow layer of the PRAN is that has been combined horizontally, and PRAN is the complete network after assembling of PRA modules vertically.

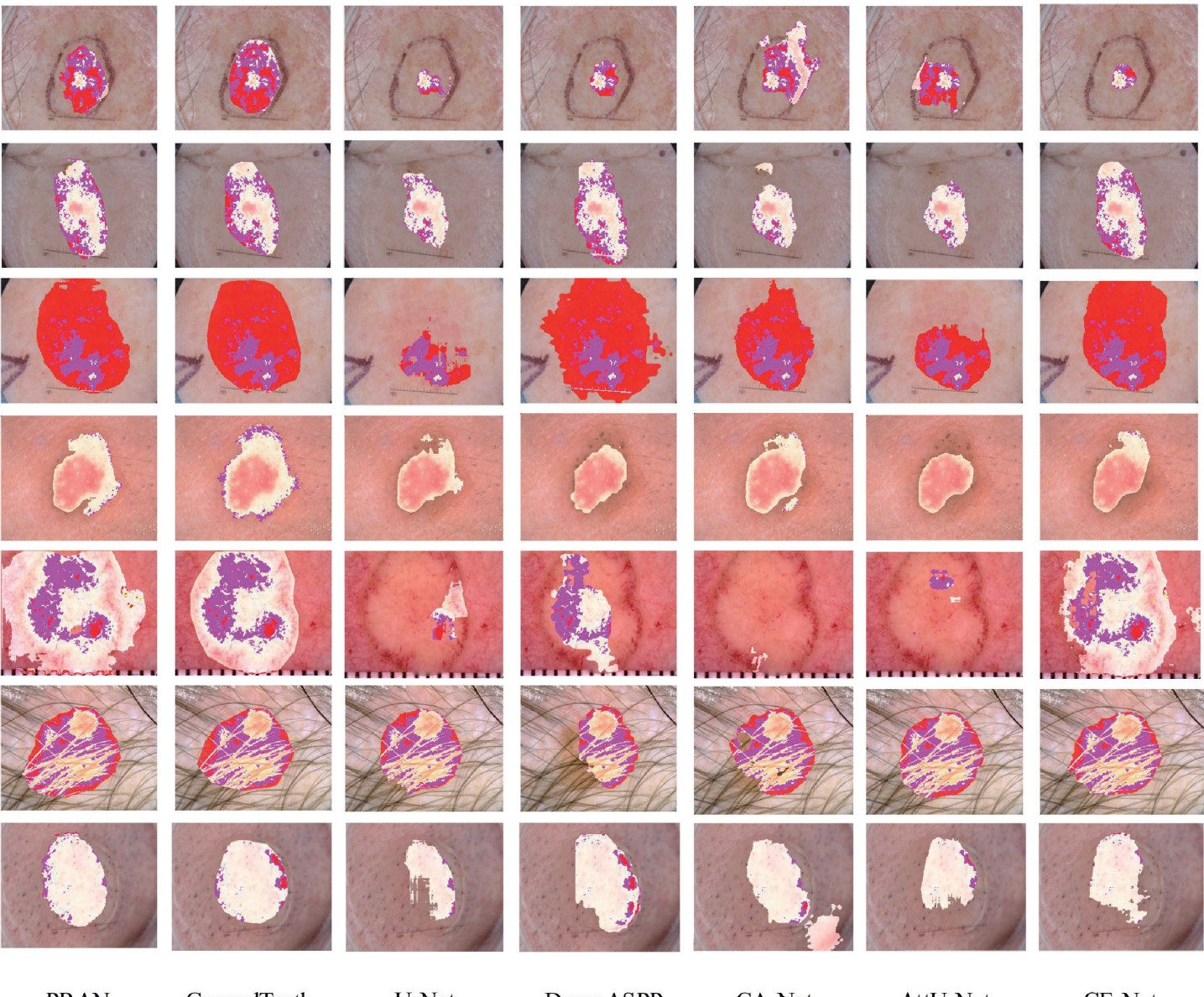

**Fig 9.** Visualization results of segmentation on ISIC2017, the discolored area represents the segmentation result of the model or label, and the darker part of the segmented lesion area indicates the part of the original image where the skin color is obvious.

## 4.5. The generalization ability

To evaluate the generalization ability of the proposed method, we assessed it on another challenge. We used the KvasirSEG [58] dataset to evaluate our model. In the above experimental setting, we initialized the images of different sizes to 384×384 size. Because KvasirSEG dataset consists of 1000 images, we randomly disorganized them and took the 500s as the training set, 100s as the validation set, and 400s as the test set. We used the same data enhancement method as the ISIC dataset. Fig 11 and Table 6 show the experimental results and specific metrics. It shows that the addition of attention affected the segmentation performance of the model, and in some cases, it even increased the learning burden of the model significantly. For example,

**Table 4. Comparison of PRAN with existing models on ISIC2017 dataset. Bold data in the table indicates the highest metric in the column.**

| Network | year | DC(%) | MIOU(%) | SP(%) | ACC(%) | SE(%) | Para(M) |
|---|---|---|---|---|---|---|---|
| U-Net [8] | 2015 | 84.39±8.42 | 77.30±11.29 | 90.06±5.94 | 90.65±5.96 | 91.28±5.98 | 31 |
| Attention U-Net [9] | 2018 | 86.96±6.98 | 80.37±9.80 | 91.20±6.17 | 91.80±6.19 | 92.43±6.23 | 33.3 |
| DenseASPP [52] | 2018 | 87.27±7.81 | 80.45±7.81 | 91.71±5.31 | 92.32±5.30 | 92.96±5.31 | 33.8 |
| CE-Net [51] | 2019 | 88.73±7.21 | 81.23±9.98 | 92.16±5.37 | 92.81±5.36 | 93.49±5.36 | 24 |
| CA-Net [11] | 2020 | 88.17±7.81 | 80.24±10.75 | 91.72±5.76 | 92.37±5.78 | 93.29±5.81 | 2.8 |
| Abhishek et al. [43] | 2020 | 83.86±7.80 | 75.1±10.75 | 95.16±3.7 | 92.20±4.5 | 87.06±7.7 | - |
| DAGAN [57] | 2020 | 85.9 | 77.1 | **97.6** | 93.5 | 83.5 | - |
| EUnet-DGF [23] | 2020 | 87.89 | 80.37 | - | 94.77 | - | - |
| DS-Net [50] | 2020 | 87.5 | 77.5 | - | 95.5 | - | - |
| Scale-Att-ASPP [7] | 2020 | 87.81 | 80.28 | 93.20 | 93.16 | 86.97 | - |
| FC-DPN [53] | 2020 | 84.56 | 76.34 | 93.71 | **98.65** | 83.82 | - |
| **PRAN(proposed)** | 2021 | **89.47±6.31** | **82.14±9.20** | 92.51±4.65 | 93.17±4.62 | **93.86±4.61** | 21.6 |

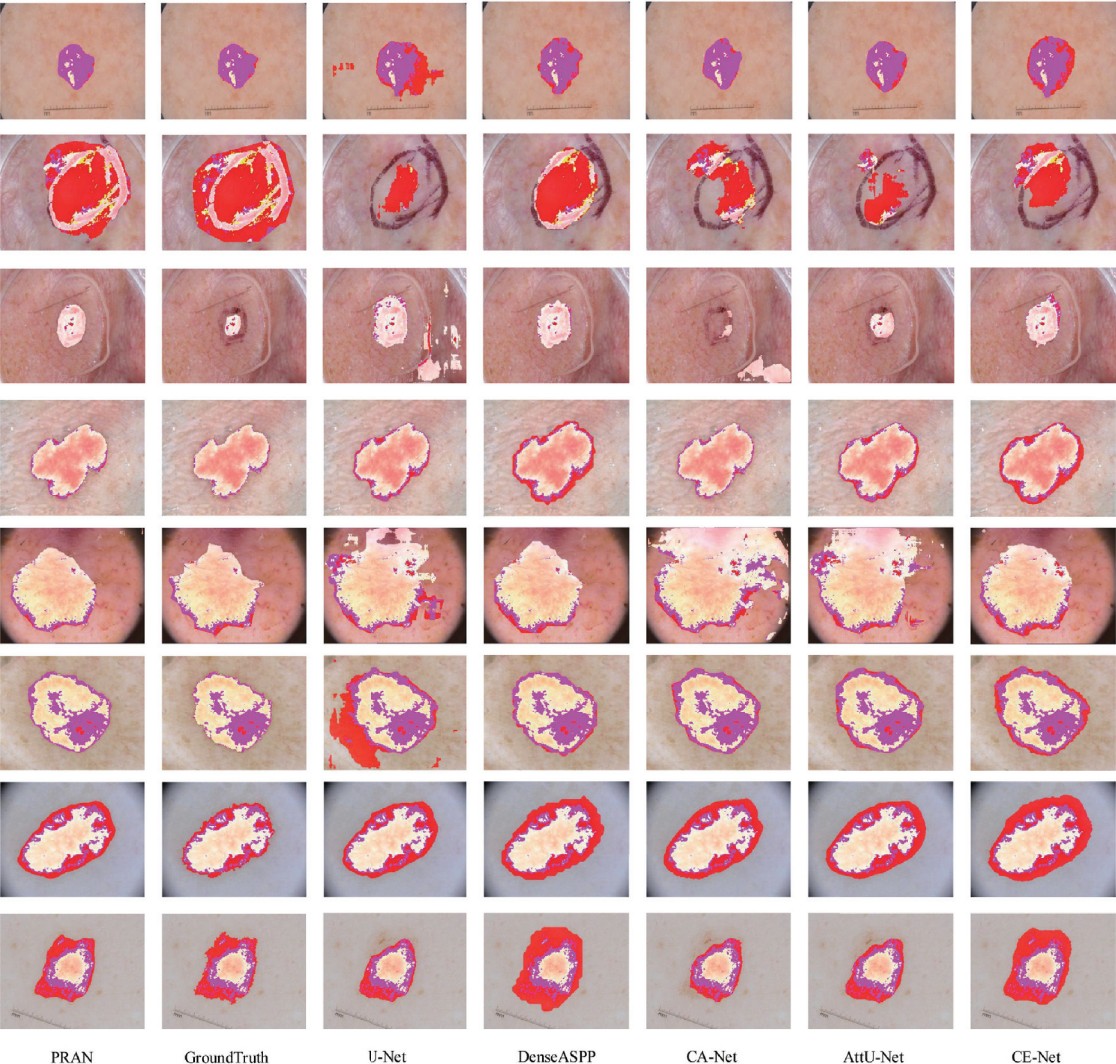

**Fig 10.** Visualization results of segmentation on ISIC2018, the discolored part represents the segmentation result of the lesion area on the original image, and the darker part of the segmented lesion area indicates the part of the original image where the skin color is obviously.

**Table 5. Comparison table of segmentation performance on the ISIC2018 dataset.**

| Network | year | DC(%) | MIOU(%) | SP(%) | ACC(%) | SE(%) | Para(M) |
|---|---|---|---|---|---|---|---|
| U-Net [8] | 2015 | 88.43±5.15 | 82.25±7.56 | 93.05±3.33 | 93.59±3.33 | 94.15±3.33 | 31 |
| SLSDeep [39] | 2018 | 89.4 | 83.2 | 93.3 | 93.7 | 90.4 | - |
| Attention U-Net [9] | 2018 | 90.95±5.11 | 85.51±7.51 | 94.45±3.12 | 95.01±3.12 | 95.58±3.13 | 33.3 |
| DenseASPP [52] | 2018 | 91.40±5.44 | 84.88±7.87 | 93.60±3.17 | 94.16±3.18 | 94.74±3.19 | 33.8 |
| BCDU-Net [54] | 2019 | 82.24 | - | 97.86 | 95.60 | 80.07 | - |
| CE-Net [51] | 2019 | 89.8±5.39 | 83.1±7.82 | 93.4±3.21 | 94.1±3.21 | 93.4±3.22 | 24 |
| DA-Net [55] | 2019 | 89.6 | 82.8 | 93.7 | 94.0 | 93.7 | - |
| ADAM [12] | 2019 | 90.8 | 84.4 | 94.1 | 94.7 | 94.2 | - |
| CA-Net [11] | 2020 | 92.68±5.78 | 87.10±8.48 | 94.76±3.71 | 95.33±3.70 | 95.92±3.73 | 2.7 |
| DAGAN [57] | 2020 | 88.5 | 82.4 | 91.1 | 92.9 | 95.3 | - |
| DEDB [56] | 2021 | 90.00 | 83.30 | **97.00** | **96.95** | **96.50** | - |
| **PRAN(Proposed)** | 2021 | **93.37±4.50** | **87.94±6.68** | 94.98±2.98 | 95.56±2.99 | 96.15±2.97 | 21.6 |

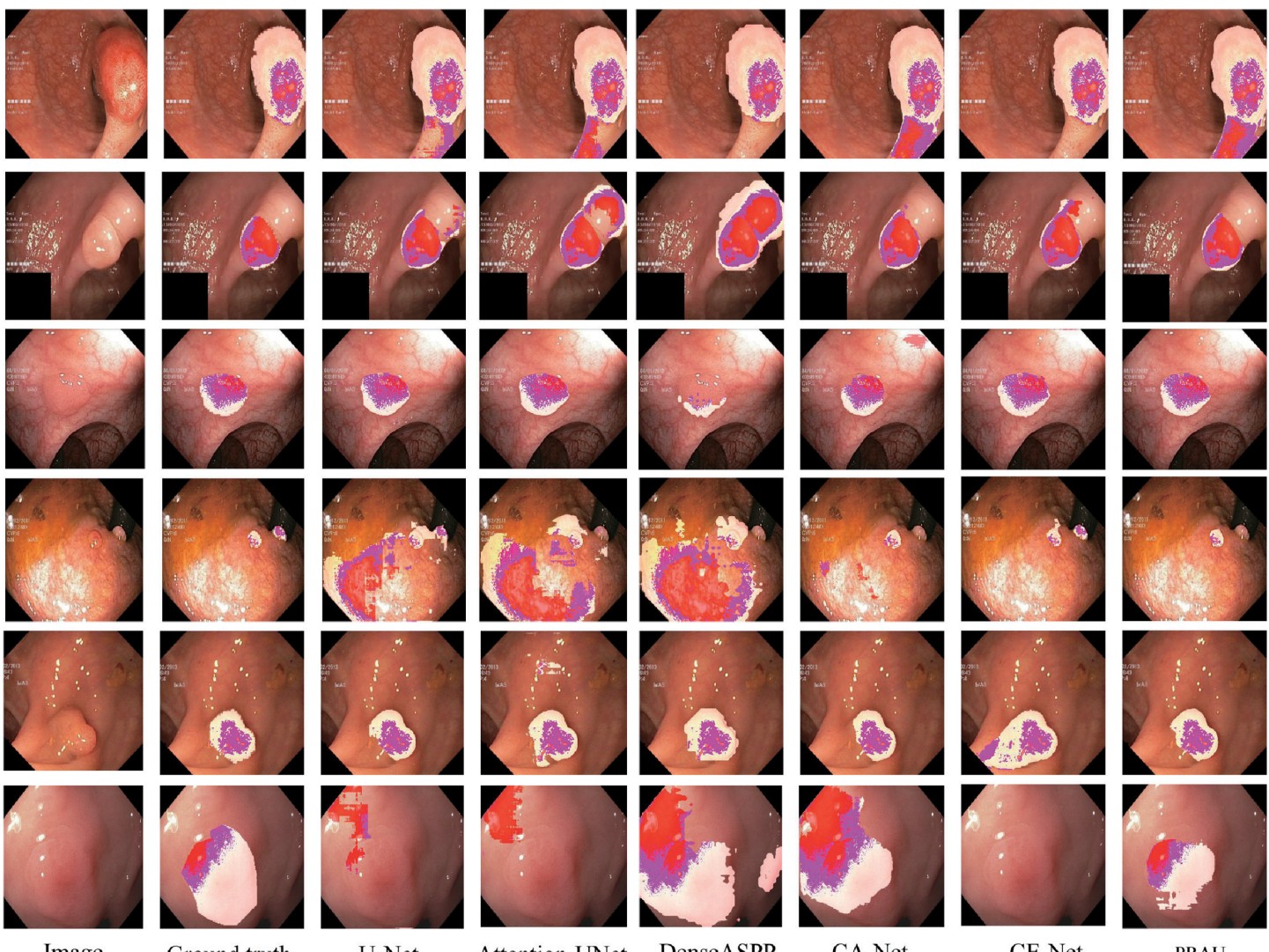

**Fig 11. Visualization results of segmentation on KvasirSEG dataset.** The discolored part represents the segmentation result of the lesion area on the original image. The darker part of the segmented lesion area indicates the part of the original image where the skin color is clear.

**Table 6. Comparison test table of generalizability of the proposed model on KvasirSEG dataset.** Bold data in the table indicates the highest metric in the column.

| Network | year | DC(%) | MIOU(%) | SP(%) | ACC(%) | SE(%) | Para(M) |
|---------|------|-------|---------|-------|--------|-------|---------|
| U-Net [8] | 2015 | 89.40±5.33 | 82.12±6.64 | 94.01±3.48 | 94.66±3.47 | 95.94±3.46 | 31 |
| Attention U-Net [9] | 2018 | 89.16±5.71 | 81.81±8.30 | 93.75±3.45 | 94.40±3.44 | 95.08±3.44 | 33.3 |
| DenseASPP [52] | 2018 | 89.63±6.64 | 82.76±9.33 | 94.25±3.84 | 94.91±3.84 | 95.59±3.85 | 33.8 |
| CE-Net [52] | 2019 | 89.95±5.72 | 83.07±8.32 | 94.35±3.24 | 95.01±3.21 | 95.69±3.20 | 24 |
| ADAM [12] | 2019 | 88.22 | 81.37 | **96.68** | **98.28** | 91.04 | - |
| CA-Net [11] | 2020 | 88.55±6.19 | 80.99±8.65 | 93.57±3.77 | 94.22±3.76 | 94.89±3.76 | 2.7 |
| **PRAU(Proposed)** | 2021 | **90.88±5.39** | **84.41±8.01** | 94.73±3.46 | 95.40±3.43 | **96.09±3.41** | 32.7 |

CA-Net made the model not as effective as it should be. Replacing the bottom module in U-Net with PRA can improve the model performance. The improvement of the segmentation of PRAU is in the figure, which is significantly more than several other models.

## 5. Conclusion

In this paper, we propose a novel Pyramid Residual Attention Network(PRAN) which shows the advantages of the encoder-decoder idea from a new perspective and apply it successfully to the skin lesion image segmentation tasks. The PRA module can effectively extract the information of the feature map, which alleviates the information confusion problem caused by the feature pyramid to a certain extent. Starting from the PRA module, we further design a new meaning to solve the inadequate feature learning in traditional neural networks. Using channel attention as the supervising mechanism of the feature pyramid, the model ensures accurate segmentation of dermoscopy images with distinctive features and determines the approximate area of image lesions with high interference and low contrast. Importantly, our model achieves satisfactory results on both ISIC2017 dataset and ISIC2018 dataset. The base module has room for further improvement, which is still of research interest for other types of edge segmentation tasks.

## Author Contributions

**Conceptualization:** Jing Liang.

**Investigation:** Xin Lin.

**Methodology:** Jinkun Dong.

**Project administration:** Yun Jiang.

**Resources:** Tongtong Cheng.

**Validation:** Jing Liang.

**Visualization:** Yuan Zhang.

**Writing – original draft:** Tongtong Cheng.

**Writing – review & editing:** Huixia Yao.

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
