## [Decision Letter · Decision Letter 0]

24 Nov 2021

PONE-D-21-32574Dermoscopic Image Segmentation Based on Pyramid Residual Attention Module.PLOS ONE

Dear Dr. Cheng,

Thank you for submitting your manuscript to PLOS ONE. After careful consideration, we feel that it has merit but does not fully meet PLOS ONE’s publication criteria as it currently stands. Therefore, we invite you to submit a revised version of the manuscript that addresses the points raised during the review process.

We look forward to receiving your revised manuscript.

Kind regards,

Sen Xiang

Academic Editor

PLOS ONE

Journal Requirements:

“No.”

5. Please ensure that you refer to Figure 5 in your text as, if accepted, production will need this reference to link the reader to the figure.

Reviewers' comments:

Reviewer's Responses to Questions

**Comments to the Author**

1. Is the manuscript technically sound, and do the data support the conclusions?

Reviewer #1: Yes

Reviewer #2: Yes

Reviewer #3: Yes

Reviewer #4: Yes

2. Has the statistical analysis been performed appropriately and rigorously? 

Reviewer #1: N/A

Reviewer #2: Yes

Reviewer #3: Yes

Reviewer #4: No

3. Have the authors made all data underlying the findings in their manuscript fully available?

Reviewer #1: Yes

Reviewer #2: Yes

Reviewer #3: Yes

Reviewer #4: No

4. Is the manuscript presented in an intelligible fashion and written in standard English?

Reviewer #1: Yes

Reviewer #2: Yes

Reviewer #3: Yes

Reviewer #4: No

5. Review Comments to the Author

Reviewer #1: The authors propose a new pyramidal residual attention(PRA) Network for dermoscopic image segmentation. The authors validated the method on two public available datasets - ISIC 2017 and 2018. Experiments showed the proposed method outperforms all state of the art methods. Ablation study also confirmed the contribution of each module.

Concerns:

1. Some arguments are not supported, especially the limitation on attention mechanism (Line 48). These limitations are not easy to understand and not well supported. The authors either need to demonstrate these limitations, or at least add reference describing these limitations.

2. Claims, such as "the PRA Network is more interpretable." and "... to ensure the stability and robustness of the model" (Line 69) are not supported by the experiment. The authors need to add experiment/visualization to support these claims. Otherwise, the authors may want to remove these unsupported claims.

3. The authors did not provide variance of the reported results, given the DC/mIoU improvement over the CA-Net [11] is small (0.7%). The authors may want to add variance measurement to these scores, or use cross validation.

4. There are many typos and grammar errors in the manuscript. The authors need to greatly improve their writings. These are not limited to the following:

- Line 21, "nature language processing" -> "natural language processing"

- Line 153, capital M

- Line 354, "Table reftable4"

- Line 61-62, the sentence is difficult to understand.

Reviewer #2: This paper proposes a novel feature extraction module PRA(Pyramid Residual Attention) for dermoscopic image segmentation. In detail, PRA consists of a residual module, an

attention module, and a pyramid module. PRA makes use of the denoising function of the encoder-decoder structure and combines it with a multi-scale feature pyramid. The

channel attention is used to monitor the feature extraction process of the pyramid to

ensure the efficiency of the PRA. Experiments on ISIC2017 and ISIC2018 show the efficiency of the proposed model.

There are major weaknesses in this paper that should be solved:

1. Figure 2 and Figure 3 are so messy that the reader cannot understand the architecture of the proposed module. Please use a more concise figure to illustrate the PRA network.

2. There are four identical convolution modules above each feature pyramid, and I hope the authors explain what they do.

3. There are many errors in the paper, please check the paper in detail by the authors. For example, there are missing spaces at the end of the sentences in lines 217 and 241.

4. All experiments in this paper were done on the ISIC dataset. Please conduct the same experiments on other datasets to verify the generalization ability of the proposed model. The explanation for the experiment is inadequate. More analysis should be added to demonstrate the core idea of the whole paper.

5. Some missing key references about segmentation, and residual attention [1,2,3,4] as these methods have been widely used in these works. [1] Deep Object Tracking with Shrinkage Loss, ieee tpami

[2] See more, know more: Unsupervised video object segmentation with co-attention siamese networks, cvpr

[3] Zero-shot video object segmentation with co-attention siamese networks, ieee tpami

[4] Segmenting Objects from Relational Visual Data, ieee tpami

6. The method proposed in this paper is for dermoscopic images. What is the difference between the proposed model and the common image segmentation method? Is the proposed model effective for everyday image segmentation?

Reviewer #3: 1. The language needs to be further enhanced. There are still many difficult sentences and case errors in the current manuscript.

2. The author proposes a new attentional mechanism for the skin lesions segmentation. Some existing studies have been carried out on attentional network segmentation, such as "Lightweight attention convolutional neural network for retinal vessel image segmentation ". The novelty of the attentional mechanism proposed by the authors requires further illustration.

Reviewer #4: This manuscript proposes a Pyramid Residual Attention Module and the corresponding Pyramid Residual Attention network. My major concern is the novelty and presentation.

i) The Pyramid Attention mechanism is an existing method that is proposed in [1]. In this work, a module named Pyramid Residual Attention is proposed. However, after going through section 3.2, I find that the proposed method is just a combination of Residual and pyramid attention. Unfortunately, the original work [1] used the residual block as well.

ii) One of the targeting challenges in this manuscript is s the multi-scale problem. However, there is an existing work that uses pyramid attention for multi-scale image fusion [2]. It was published on Apr 2021 which is earlier than the submission date.

iii) What is the feature pyramid exactly? The output of the Pyramid residual attention? More justifications are required.

iv) This work focuses on the segmentation and the attention mechanism, why does GAN matter? I cannot see anything proposed in this work is related to the generative model.

v) Important baseline is missing. Check [3]. Moreover, existing models that use pyramid attention should be compared. For example, [2]. It is not designed for Dermoscopic Image Segmentation but is still worth doing as dermoscopic image belongs to the medical image.

vi) The results are not statistically significant which lacks the std etc.

vii) Section 3 should be "Methods" or "Methodology". Authors need to double-check the whole manuscript to fix the typos.

[1]Mei, Y., Fan, Y., Zhang, Y., Yu, J., Zhou, Y., Liu, D., Fu, Y., Huang, T.S. and Shi, H., 2020. Pyramid attention networks for image restoration. arXiv preprint arXiv:2004.13824.

[2]Fu, J., Li, W., Du, J. and Huang, Y., 2021. A multiscale residual pyramid attention network for medical image fusion. Biomedical Signal Processing and Control, 66, p.102488.

[3] Abhishek, K., Hamarneh, G. and Drew, M.S., 2020. Illumination-based transformations improve skin lesion segmentation in dermoscopic images. In Proceedings of the IEEE/CVF Conference on Computer Vision and Pattern Recognition Workshops (pp. 728-729).

6. PLOS authors have the option to publish the peer review history of their article (what does this mean?). If published, this will include your full peer review and any attached files.

Reviewer #1: **Yes: **Hao Tang

Reviewer #2: No

Reviewer #3: No

Reviewer #4: No

---

## [Author Response · Author response to Decision Letter 0]

3 Jan 2022

[Answers for Reviewer 1’ Comments]

First of all, we would lie to thank you for all that you commented for our paper.

Point 1: Some arguments are not supported, especially the limitation on attention mechanism (Line 48). These limitations are not easy to understand and not well supported. The authors either need to demonstrate these limitations, or at least add reference describing these limitations.

Response 1:

 CA-Net placed the attention module on the up-sampling and skip connection based on U-Net to improve the performance of the network greatly. An adaptive bi-directional attention module was proposed in ADAM which supplements the shallow extraction results with the attention information at the bottom level, thereby to process information of different layers effectively.

Although the above attention mechanisms can improve the segmentation effect of neural networks to different degrees, they most contain the following:

 1.The single convolutional kernel has limited ability to learn diverse features, and the various size of the receptive field has a large impact on the feature extraction process. The multi-scale feature fusion mechanism should be used to enhance the feature extraction ability of the model for feature-diverse data sets.

2.The importance of the convolutional layer has been ignored for each stage during the design of the network framework. It is reasonable to adopt various convolutional strategies when confronted with different sizes of semantic information with different densities. Deeper sub-networks could be assigned to sufficiently extract messages in the shallow stage of convolution with comprehensive information.

3.The process of feature extraction in classic networks does not screen the information, ignoring the different situations of the amount of information extracted by the convolutional kernels in the feature extraction module. The module which is responsible for extracting information should be supervised and managed accordingly, so as to enhance the update iteration of important convolutional kernels and reduce the unnecessary training cost.

 We are grateful for this suggestion. In this sentence, I would like to express that attention mechanism could alleviate these problem of conventional CNN but the three problems exist still. 

Point 2: Claims, such as "the PRA Network is more interpretable." and "... to ensure the stability and robustness of the model" (Line 69) are not supported by the experiment. The authors need to add experiment/visualization to support these claims. Otherwise, the authors may want to remove these unsupported claims.

Response 2:

As shown in Fig 8, the loss decline curves for training and validation, our proposed PRA model is able to reach a good level of loss in 50 iterations during training and validation, and makes a small and stable decline in the training loss in the last 250 iterations. Moreover, our PRA Network has a more obvious advantage over UNet in terms of loss reduction and is more stable.

We are grateful for this suggestion. In this regard, we decide to delete these not rigorous sentences, Because we lack of these work to prove the interpretability. And in order to valid the stability of our model, we supply the loss record in training and validation.

Point 3: The authors did not provide variance of the reported results, given the DC/mIoU improvement over the CA-Net [11] is small (0.7%). The authors may want to add variance measurement to these scores, or use cross validation.

Response 3:

Network year DC MIOU SP ACC SE Para(M)

U-Net [8] 2015 84.39±8.42 77.30±11.29 90.06±5.94 90.65±5.96 91.28±5.98 31

ATTU-Net[9] 2018 86.96±6.98 80.37±9.80 91.20±6.17 91.80±6.19 92.43±6.23 33.3

DenseASPP[53] 2018 87.27±7.81 80.45±7.81 91.71±5.31 92.32±5.30 92.96±5.31 33.8

CE-Net[52] 2019 88.73±7.21 81.23±9.98 92.16±5.37 92.81±5.36 93.49±5.36 24

CA-Net[11] 2020 88.17±7.81 80.24±10.75 91.72±5.76 92.37±5.78 93.29±5.81 2.8

DAGAN[58] 2020 85.9 77.1 97.6 93.5 83.5 -

EUnet-DGF[23] 2020 87.89 80.37 - 94.77 - -

DS-Net[51] 2020 87.5 77.5 - 95.5 - -

Scale-Att-ASPP [7] 2020 87.81 80.28 93.20 93.16 86.97 -

 FC-DPN[54] 2020 84.56 76.34 93.71 98.65 83.82 -

PRA Network(proposed) 2021 89.47±6.31 82.14±9.20 92.51±4.65 93.17±4.62 93.86±4.61 21.6

We are grateful for this suggestion. In this regard, we add the std to our result tables. Because the models in this segmentation task improve not obvious, although one model could segment the lesion in high contrast images easily, if encountering the low contrasts, it will perform bad. So in the table, we can see that the result of model is higher litter than other models’, but it can be well when apply to the task.

Point 4: There are many typos and grammar errors in the manuscript. The authors need to greatly improve their writings.

Response 3:

We are grateful for this suggestion. In this regard, we carefully checked and corrected the mistakes.

Thanks for your comments.

[Answers for Reviewer 2’ Comments]

First of all, we would lie to thank you for all that you commented for our paper.

Point 1: Figure 2 and Figure 3 are so messy that the reader cannot understand the architecture of the proposed module. Please use a more concise figure to illustrate the PRA network.

Response 1:

We are grateful for this suggestion. 

In this regard, we change the figure 2 to this appearance. And the figure 3 is our designed to present the process of PRA from accepting the feature to get the output. The detail is narrated at the 3.2.

We propose a streamlined and effective synthesis module called the pyramidal residual attention (PRA) module. To better understand the principle of the module's composition, we use a simple convolutional neural network model of feature learning concept to illustrate. Fig3 shows the flow of the PRA module for processing feature maps. Taking the deep level of the network as an example, the size of the input image and the output feature map is 16×20. Firstly, the features are processed on the basis of the original map using 32 convolution kernels size of 3×3 as a result of this level of information extraction. At the same time, the input feature maps are compressed by down-sampling once, using a 2×2 pooling layer with a step size of 2, concentrating the semantic information and ignoring the useless spatial information. The output is fed into a feature pyramid to learn the feature information got from different sizes of receptive fields. The feature sets of multiple receivers are used as the input for the channel attention. Channel attention takes the results of different convolutional blocks to derive the information weight of each channel through a pooling layer. Thus these options can enhance the input to different degrees to focus the attention of the model. Finally, after recovered the size by the deconvolution layer, the feature maps connect with the output of the residual block to obtain the output of the module.

Point 2: There are four identical convolution modules above each feature pyramid, and I hope the authors explain what they do.

Response 2:

We are grateful for this suggestion. 

The four convolution is our feature pyramid. It is used to study the different knowledge from different convolution. Because convolution has their own perceptual fields. The details like following:

Feature pyramids are feature extractors that incorporate different kinds of convolutional kernels. Because the convolution operation is pixel-by-pixel, features of different sizes can be learned efficiently by these convolution kernels. Four convolutional kernels of 226 1×1, 3×3, 5×5 and 7×7 are used to process the input in parallel. Multiple different convolutional kernels are able to learn different knowledge from different perceptual fields [45], adding more semantic and spatial information to the results of the original 229 single 3×3 convolutional kernel. The stride of the convolution kernel is set to 1, which 230 does not change the size of the image while feature extraction, so that the features of the image can be substantially preserved and provide information security for subsequent learning. 

Point 3: There are many errors in the paper, please check the paper in detail by the authors. For example, there are missing spaces at the end of the sentences in lines 217 and 241.

Response 3:

We are grateful for this suggestion. 

In this regard, we carefully checked and corrected the mistakes.

Point 4: All experiments in this paper were done on the ISIC dataset. Please conduct the same experiments on other datasets to verify the generalization ability of the proposed model. The explanation for the experiment is inadequate. More analysis should be added to demonstrate the core idea of the whole paper.

Response 4:

We are grateful for this suggestion. 

In this regard, we add the generative experiment of KvasirSEG datatset to valid our model. The detail is following.

Network year DC MIOU SP ACC SE Para(M)

U-Net[8] 2015 89.40±5.33 82.12±6.64 94.01±3.48 94.66±3.47 95.94±3.46 31

AttU-Net[9] 2018 89.16±5.71 81.81±8.30 93.75±3.45 94.40±3.44 95.08±3.44 33.3

DenseASPP[53] 2018 89.63±6.64 82.76±9.33 94.25±3.84 94.91±3.84 95.59±3.85 33.8

CE-Net[53] 2019 89.95±5.72 83.07±8.32 94.35±3.24 95.01±3.21 95.69±3.20 24

ADAM[12] 2019 88.22 81.37 96.68 98.28 91.04 -

CA-Net[11] 2020 88.55±6.19 80.99±8.65 93.57±3.77 94.22±3.76 94.89±3.76 2.7

PRAU(proposed) 2021 90.88±5.39 84.41±8.01 94.73±3.46 95.40±3.43 96.09±3.41 32

We use the KvasirSEG dataset to evaluate our model. in the above experimental setting, we initialize the images of different sizes to 384×384 size. the KvasirSEG dataset consists of 1000 images, we randomly disorganize them and take 500 as the training set, 100 as the validation set, and finally 400 as the test set. Using the same data enhancement method as the above dataset, the effect is shown in Fig. The specific metrics are shown in Table. It is shown in the experiments that the addition of attention affects the segmentation performance of the model, and in some cases, it even increases the learning burden of the model significantly. For example, CANet, after using attention instead, makes the model not as effective as it should be. Combining PRA with UNet and using the PRA module to replace the bottom module in UNet can improve the model performance. The improvement of the segmentation effect of UPRA can be clearly felt from the figure, which is significantly more than several other models.

Point 5: Some missing key references about segmentation, and residual attention [1,2,3,4] as these methods have been widely used in these works. 

Response 5:

We are grateful for this suggestion.

In this regard, we The following papers are cited as a supplement.

1.Lu et al.[26] proposed a CO-attention mechanism which can enhance the model to capture remote information.

2.Lu et al.[40] proposed the Shrinkage loss function to balance number of data of the different classes.

Thanks for your comments.

[Answers for Reviewer 3’ Comments]

First of all, we would lie to thank you for all that you commented for our paper.

Point 1: The language needs to be further enhanced. There are still many difficult sentences and case errors in the current manuscript.

Response 1:

We are grateful for this suggestion. In this regard, we carefully checked and corrected the mistakes.

Point 2: The author proposes a new attentional mechanism for the skin lesions segmentation. Some existing studies have been carried out on attentional network segmentation, such as "Lightweight attention convolutional neural network for retinal vessel image segmentation ". The novelty of the attentional mechanism proposed by the authors requires further illustration.

Response 2:

We are grateful for this suggestion. In this regard, we want to interpret as following.

In this paper, we want to introduce a big base block, and connect itself to self. And to do this need a condition that the base block need to have a good study ability. So we use the pyramid, attention and residual for the litter module. And fortunately, this thought get a good result in the task of skin lesion segmentation. The attetion is not attention but a base block. 

Thanks for your comments.

[Answers for Reviewer 4’ Comments]

First of all, we would lie to thank you for all that you commented for our paper.

Point 1: The Pyramid Attention mechanism is an existing method that is proposed in [1]. In this work, a module named Pyramid Residual Attention is proposed. However, after going through section 3.2, I find that the proposed method is just a combination of Residual and pyramid attention. Unfortunately, the original work [1] used the residual block as well.

Response 1:

We are grateful for this suggestion. In this regard, we carefully read the paper[1], and find that model was combining the different convolution kernel and scale agnostic attention, and the pyramid will reduce the feature size, importantly, the pyramid attention is consist of pyramid and SE-block-liker. This work is the same as the [7] Wei Z, Shi F, Song H, et al. Attentive boundary aware network for multi-scale skin lesion segmentation with adversarial training[J]. Multimedia Tools and Applications, 2020, 79(37): 27115-27136.

Otherwise, the PRA in our work is not only a attention mechanism, but it is also a base block for PRA Network. 

2.2 Residual and Pyramid Attention Networks(RPAN)

The pyramid structure and the residual structure are identical to attention in a variety of ways, and they are all capable of improving the learning ability of the model to varying degrees. Mei et al.[28] proposed a pyramidal attention structure for image recovery tasks, combining convolution kernels of different sizes with scale agnostic attention to learn the global feature information of an image. Fu et al.[29] proposed a residual pyramid attention network for CT image segmentation by combining the inception-like module with SE-block and attention block consist of a small encoder-decoder, and then combining the two to form a feature extraction module with a large number of parameters. Chae et al.[30] proposed a residual UNet network combined with SE-block for wound region segmentation, and a modified version of SE-block was added to the skip-connected part of UNet to improve the shallow information transfer efficiency of the model. SAR-U-Net[31] proposed a combination of SE-Block and the pyramid pooling embedding in the Res-UNet. This composition could improve the ability of feature capture of the down sampling in ResUNet. Shah et al.[32] proposed the use of Astrous convolution and residual structure for the enhancement of U-Net models, using residual and Astrous convolution as the base convolution unit to improve the learning ability of the model on features. Flaute et al.[33] proposed a residual channel attention network for sampling and recovery of super-resolution images, which can well preserve the integrity of features learned from the encoder. Punn et al.[34] proposed a residual space cross-attention-guided inceptionUnet model that fuses shallow and deep semantic information and improves the extraction capability of a single convolutional block with the inception[35] structure, which improved the feature learning of the model at the base.

Point 2: One of the targeting challenges in this manuscript is s the multi-scale problem. However, there is an existing work that uses pyramid attention for multi-scale image fusion [2]. It was published on Apr 2021 which is earlier than the submission date.

Response 1:

We are grateful for this suggestion. In this regard, we carefully read this paper, and find that it may be a conflict of same name. The model in paper [2] consists of pyramid attention and residual attention, and two block is just Front back combination, and finally get the name residual pyramid attention.

Point 3:  What is the feature pyramid exactly? The output of the Pyramid residual attention? More justifications are required

Response 1:

We are grateful for this suggestion. In this regard, we elaborate the detail of feature pyramid. It is just a convolution module which consists of four convolution kernels 1x1, 3x3, 5x5, 7x7. these convolution kernel could study different knowledge from the same feature map. So, pyramid just likes different perception for a same feature map.

The output of Pyramid residual attention is a weighted feature map. And it consists of the original input and the feature map throught pyramid and attention mechanism. Weighted feature map will be screened in the layers com after PRA.

Point 4: This work focuses on the segmentation and the attention mechanism, why does GAN matter? I cannot see anything proposed in this work is related to the generative model.

Response 1:

We are grateful for this suggestion. In this regard, we change 2.2 to Residual and Pyramid Attention Networks(RPAN).

Point 5:  Important baseline is missing. Check [3]. Moreover, existing models that use pyramid attention should be compared. For example, [2]. It is not designed for Dermoscopic Image Segmentation but is still worth doing as dermoscopic image belongs to the medical image.

Response 1:

We are grateful for this suggestion. In this regard, we add the result of paper [3] to our table. 

Point 6: The results are not statistically significant which lacks the std etc.

Response 6:

We are grateful for this suggestion. In this regard, we add the std in our result tables.

Network year DC MIOU SP ACC SE Para(M)

U-Net [8] 2015 84.39±8.42 77.30±11.29 90.06±5.94 90.65±5.96 91.28±5.98 31

ATTU-Net[9] 2018 86.96±6.98 80.37±9.80 91.20±6.17 91.80±6.19 92.43±6.23 33.3

DenseASPP[53] 2018 87.27±7.81 80.45±7.81 91.71±5.31 92.32±5.30 92.96±5.31 33.8

CE-Net[52] 2019 88.73±7.21 81.23±9.98 92.16±5.37 92.81±5.36 93.49±5.36 24

CA-Net[11] 2020 88.17±7.81 80.24±10.75 91.72±5.76 92.37±5.78 93.29±5.81 2.8

DAGAN[58] 2020 85.9 77.1 97.6 93.5 83.5 -

EUnet-DGF[23] 2020 87.89 80.37 - 94.77 - -

DS-Net[51] 2020 87.5 77.5 - 95.5 - -

Scale-Att-ASPP [7] 2020 87.81 80.28 93.20 93.16 86.97 -

 FC-DPN[54] 2020 84.56 76.34 93.71 98.65 83.82 -

PRA Network(proposed) 2021 89.47±6.31 82.14±9.20 92.51±4.65 93.17±4.62 93.86±4.61 21.6

Point 7: Section 3 should be "Methods" or "Methodology". Authors need to double-check the whole manuscript to fix the typos.

Response 7:

We are grateful for this suggestion. In this regard, we carefully checked and corrected the mistakes.

[1]Mei, Y., Fan, Y., Zhang, Y., Yu, J., Zhou, Y., Liu, D., Fu, Y., Huang, T.S. and Shi, H., 2020. Pyramid attention networks for image restoration. arXiv preprint arXiv:2004.13824.

[2]Fu, J., Li, W., Du, J. and Huang, Y., 2021. A multiscale residual pyramid attention network for medical image fusion. Biomedical Signal Processing and Control, 66, p.102488.

[3] Abhishek, K., Hamarneh, G. and Drew, M.S., 2020. Illumination-based transformations improve skin lesion segmentation in dermoscopic images. In Proceedings of the IEEE/CVF Conference on Computer Vision and Pattern Recognition Workshops (pp. 728-729).

Thanks for your comments.

---

## [Decision Letter · Decision Letter 1]

15 Feb 2022

PONE-D-21-32574R1Dermoscopic Image Segmentation Based on Pyramid Residual Attention Module.PLOS ONE

Dear Dr. Cheng,

Thank you for submitting your manuscript to PLOS ONE. After careful consideration, we feel that it has merit but does not fully meet PLOS ONE’s publication criteria as it currently stands. Therefore, we invite you to submit a revised version of the manuscript that addresses the points raised during the review process.

Please follow the reviewers' comments and improve the writting and the abstract.

We look forward to receiving your revised manuscript.

Kind regards,

Sen Xiang

Academic Editor

PLOS ONE

Journal Requirements:

Reviewers' comments:

Reviewer's Responses to Questions

**Comments to the Author**

1. If the authors have adequately addressed your comments raised in a previous round of review and you feel that this manuscript is now acceptable for publication, you may indicate that here to bypass the “Comments to the Author” section, enter your conflict of interest statement in the “Confidential to Editor” section, and submit your "Accept" recommendation.

Reviewer #1: All comments have been addressed

Reviewer #2: All comments have been addressed

Reviewer #3: All comments have been addressed

2. Is the manuscript technically sound, and do the data support the conclusions?

Reviewer #1: Partly

Reviewer #2: Yes

Reviewer #3: Yes

3. Has the statistical analysis been performed appropriately and rigorously? 

Reviewer #1: N/A

Reviewer #2: Yes

Reviewer #3: Yes

4. Have the authors made all data underlying the findings in their manuscript fully available?

Reviewer #1: Yes

Reviewer #2: Yes

Reviewer #3: Yes

5. Is the manuscript presented in an intelligible fashion and written in standard English?

Reviewer #1: No

Reviewer #2: Yes

Reviewer #3: Yes

6. Review Comments to the Author

Reviewer #1: The authors have done a good job in addressing my comments. There are still some grammar mistakes, and the authors should further improve their writing, especially the abstract.

Reviewer #2: The revised manuscript has addressed all my concerns. I have no further questions about the current version

Reviewer #3: (No Response)

7. PLOS authors have the option to publish the peer review history of their article (what does this mean?). If published, this will include your full peer review and any attached files.

Reviewer #1: No

Reviewer #2: No

Reviewer #3: No

---

## [Author Response · Author response to Decision Letter 1]

15 Mar 2022

[Answers for Reviewer 1’ Comments]

First of all, we would lie to thank you for all that you commented for our paper.

Point 1: The authors have done a good job in addressing my comments. There are still some grammar mistakes, and the authors should further improve their writing, especially the abstract.

Response 1:

These are improving of our papers, as follow:

For the first, the abstract:

The diversity and complexity of skin lesion images are the challenges of dermoscopic image segmentation. Convolutional neural networks(CNNs) have played an essential role in addressing it in recent years. In this paper, to segment this kind of image better, a novel feature extraction module PRA(Pyramid Residual Attention) is designed from a new perspective. PRA consists of a residual module, an attention module, and a pyramid module. We use the denoising function of the encoder-decoder structure and combine it with a multi-scale feature pyramid. The channel attention is to monitor the feature extraction process of the pyramid to ensure the efficiency of the PRA. The PRA Network is composed of shallow, middle, and deep layers. Both the shallows and middle layer consist of two PRA modules, and the deep layer consists of the single one. The shallow, medium, deep layers in PRA Network link by down-sampling and up-sampling to improve the segmentation effect. The network is subjected to ablation experiments on the ISIC 2018 dataset to verify the effectiveness of the PRA module and achieve better results in all metrics. Our method achieved better results for Dice coefficient (DC) indices equal to 93.37% and 89.47%, for MIOU indices up to 87.94% and 82.14% respectively for the segmentation tasks on the ISIC 2018 dataset and ISIC 2017 dataset. The PRA Network is more effective compared with existing advanced methods.

We correct the mistakes and trim the word surplus in abstract.

The second, details:

1、The dermoscopy is designed to provide the physician with high-resolution images of abnormal parts of the patient’s epidermis.

Change to

The dermoscopy provides the physician with high-resolution images of abnormal parts of the patient's epidermis.

……

We have used the grammarly to check mistake of all sentences, and correct them one by one. But many use of passive voice and objective voice are confused. We try our best to improve our writing and become better. 

We are grateful for this suggestion. In this regard, we carefully checked and corrected the mistakes.

Thanks for your comments.

---

## [Editor Report · Decision Letter 2]

23 Mar 2022

PONE-D-21-32574R2Dermoscopic Image Segmentation Based on Pyramid Residual Attention Module.PLOS ONE

Dear Dr. Cheng,

Thank you for submitting your manuscript to PLOS ONE. After careful consideration, we feel that it has merit but does not fully meet PLOS ONE’s publication criteria as it currently stands. Therefore, we invite you to submit a revised version of the manuscript that addresses the points raised during the review process.

We look forward to receiving your revised manuscript.

Kind regards,

Sen Xiang

Academic Editor

PLOS ONE

Journal Requirements:

Additional Editor Comments:

Please improve the language as the reviewer suggested.
---

## [Author Response · Author response to Decision Letter 2]

4 Apr 2022

[Answers for academic editor’ Comments]

：Please improve the language as the reviewer suggested.

Response 1:

We try our best to improve our paper, and using the grammarly to guarantee the grammar correct. In order to make the paper normal, we motivate the good paper’s rhetoric. Thanks for your comprehension and supporting.

There are changes in our paper compared to the last version.

Firstly, the abstract is changed from 

The diversity and complexity of skin lesion images are the challenges of dermoscopic image segmentation. Convolutional neural networks(CNNs) have played an essential role in addressing it in recent years. In this paper, to segment this kind of image better, a novel feature extraction module PRA(Pyramid Residual Attention) is designed from a new perspective. PRA consists of a residual module, an attention module, and a pyramid module. We use the denoising function of the encoder-decoder structure and combine it with a multi-scale feature pyramid. The channel attention is to monitor the feature extraction process of the pyramid to ensure the efficiency of the PRA. The PRA Network is composed of shallow, middle, and deep layers. Both the shallows and middle layer consist of two PRA modules, and the deep layer consists of the single one. The shallow, medium, deep layers in PRA Network link by down-sampling and up-sampling to improve the segmentation effect. The network is subjected to ablation experiments on the ISIC 2018 dataset to verify the effectiveness of the PRA module and achieve better results in all metrics. Our method achieved better results for Dice coefficient (DC) indices equal to 93.37% and 89.47%, for MIOU indices up to 87.94% and 82.14% respectively for the segmentation tasks on the ISIC 2018 dataset and ISIC 2017 dataset. The PRA Network is more effective compared with existing advanced methods.

to 

We propose a stacked convolutional neural network incorporating a novel and efficient pyramid residual attention module (PRA) for the task of automatic segmentation of dermoscopic images. Precise segmentation is a significant and challenging step for computer-aided diagnosis technology in skin lesion diagnosis and treatment. The proposed PRA has the following characteristics: Firstly, we concentrate on three widely used modules in the PRA. The purpose of the pyramid structure is to extract the feature information of the lesion area at different scales, the residual means is aimed to ensure the efficiency of model training, and the attention mechanism is used to screen effective features maps. Thanks to the PRA, our network can still obtain precise boundary information that distinguishes healthy skin from diseased areas for the blurred lesion areas. Secondly, The proposed PRA can increase the segmentation ability of a single module for lesion regions through efficient stacking. Finally, we incorporate the idea of encoder-decoder into the architecture of the overall network. Compared with the traditional networks, we divide the segmentation procedure into three levels and make the pyramid residual attention network (PRAN). The shallow layer mainly processes spatial information, the middle layer refines both spatial and semantic information, and the deep layer intensively learns semantic information. The basic module of PRAN is PRA, which is enough to ensure the efficiency of the three-layer architecture network. We extensively evaluate our method on ISIC2017 and ISIC2018 datasets. The experimental results demonstrate that PRAN can obtain segmentation performance comparable to state-of-the-art deep learning models under the same experiment environment conditions. 

Secondly, the capitalization and voice problems of the first letter of the sentence are all solved, because these correcting are so many, they will be highlighted in the track version.

Thanks for your comments.

---

## [Editor Report · Decision Letter 3]

8 Apr 2022

Dermoscopic Image Segmentation Based on Pyramid Residual Attention Module.

PONE-D-21-32574R3

Dear Dr. Cheng,

We’re pleased to inform you that your manuscript has been judged scientifically suitable for publication and will be formally accepted for publication once it meets all outstanding technical requirements.

Kind regards,

Sen Xiang

Academic Editor

PLOS ONE
---

## [Editor Report · Acceptance letter]

8 Jul 2022

PONE-D-21-32574R3 

Dermoscopic Image Segmentation Based on Pyramid Residual Attention Module 

Dear Dr. Cheng:

I'm pleased to inform you that your manuscript has been deemed suitable for publication in PLOS ONE. Congratulations! Your manuscript is now with our production department. 

Kind regards, 

on behalf of

Dr. Sen Xiang 

Academic Editor

PLOS ONE